# Mobile Application Development for Prepaid Water Meter Based on LC Sensor

**DOI:** 10.3390/s24206762

**Published:** 2024-10-21

**Authors:** Ario Kusuma Purboyo, Hanif Fakhrurroja, Dita Pramesti, Achmad Rozan Chaidir

**Affiliations:** 1School of Industrial Engineering, Telkom University, Bandung 40257, Indonesia; ariokusumapurboyo@student.telkomuniversity.ac.id (A.K.P.); ditapramesti@telkomuniversity.ac.id (D.P.); achmadrozan@student.telkomuniversity.ac.id (A.R.C.); 2Research Center for Smart Mechatronics, National Research and Innovation Agency, Bandung 40135, Indonesia

**Keywords:** prepaid water meter, LC sensor, token, Bluetooth low energy, mobile application

## Abstract

This study presents a novel low-cost and low-power prepaid water meter system that combines tokenization and LC sensors to monitor water consumption accurately with mobile application via Bluetooth Low Energy (BLE) connectivity compared to conventional meters. Water meters play a vital role in monitoring water usage in Indonesia. Postpaid billing methods that rely on manual data recording are a source of concern due to potential inaccuracies caused by human error. This study presents the development of a prepaid water meter system that integrates LC sensors, BLE connectivity, a tokenization mechanism, and a mobile application to address this issue. The system offers a cost-effective solution by utilizing BLE + Global System for Mobile (GSM) from the user’s mobile phone. Using the design thinking methodology, the mobile application for the prepaid water meter achieved a usability testing score of 80. The load testing results for the back-end server, conducted with a sample size of 515 users, revealed a back-end latency of 1.973 milliseconds and an error rate of 8.74%. Furthermore, the LC sensors integrated into the PWM device showed an average error rate of 1.33%. The power consumption during each work cycle was measured at 129 mA and each battery is expected to last six years. Overall, with simple LC sensors, this system can precisely measure water usage.

## 1. Introduction

Clean water is essential for sustaining human life, as nearly all living organisms require access to clean water. Therefore, prioritizing the management and handling of clean water is imperative. In Indonesia, a dedicated institution known as the Local Water Supply Company (PDAM) is responsible for managing and providing enough clean water of good quality at a price that is accessible to the public [1].

A water meter quantifies water consumption by continuously monitoring water flow using various sensors and measurement units [2]. During this research, water billing payments to PDAM were still based on a postpaid system, where customers settle payments at the end of their usage period. To generate the water bills, PDAM dispatches its personnel at the end of each billing cycle to record water usage and then enters it into its system. This approach consumes significant time, energy, and costs for PDAM [3]. Moreover, it poses vulnerabilities such as potential errors during the recording of water usage by PDAM personnel and inaccuracies in customer water meters, leading to financial losses for both customers and PDAM [1,4]. Customers often encounter difficulties reading water usage from conventional water meters and setting usage limits on their own [4].

Focus group discussions with PDAM Tirtawening Bandung identified several challenges, including customer payment issues, water theft, and leakage, collectively known as non-revenue water (NRW). These challenges pose a threat to the financial stability of PDAM. To address these issues, a prepaid water meter (PWM) as a retrofit was developed, which is expected to assist PDAM in mitigating these problems. This system also came with a mobile application to help PDAM customers interacting with the PWM to take actions like top-ups and check the remaining balance using Bluetooth because in Indonesia Bluetooth technology has become commonplace. 

Introducing the prepaid water meter (PWM) system will result in a transition to a prepaid billing model. The cost of water fluctuates throughout various places in Indonesia. For example, the price of 10 cubic meters (m^3^) of water in the West Bandung area is IDR 78,500. Considering the minimum wage in this district is IDR 3,492,000, and the average water usage is 161.25 L per person per day (lpd), a household consisting of four people living in a medium-sized home would consume around 645 lpd, which is equivalent to 19.35 m^3^ per month [5,6,7]. The total cost for this water consumption, including taxes and additional fees, would amount to around IDR 155,000. As a result, PDAM customers would need to put aside 4.44% of their monthly income for water bills.

This work aims to create a prototype for a prepaid water meter (PWM) device that is low-cost, low-power, accurate, and secure. The STM32L431CBT6 microcontroller powers this PWM prototype, which also features an LED display, Bluetooth Low Energy (BLE) connectivity, automated meter reading (AMR) capabilities, a solenoid valve, and a tokenization mechanism. To activate the solenoid valve, users are required to top up their balance on the PWM devices. The STM32L431CBT6 is a low-power microcontroller unit (MCU) that can execute tasks based on sensor input [8,9,10]. 

The PWM utilizes Internet of Things (IoT) technology to establish communication with a mobile application using BLE. Subsequently, the mobile application uses GSM to communicate with the back-end server. By utilizing this dual communication channel, there is no requirement for infrastructure connectivity costs as it just relies on the GSM network of the user’s phone. 

IoT technology links hardware devices such as CCTV cameras, lighting systems, air conditioners, microwaves, and speakers to the Internet, enabling remote control of the device [11]. These IoT devices utilize sensors and actuators to carry out tasks, and they can be programmed to do automatic actions without requiring human intervention [12]. In 2023, some researchers developed a web-based smart dashboard application based on integrated IoT technology to integrate conventional water meters, display water usage volume, and manage invoicing. This publication uses Antares to securely store encrypted data from water meters and decrypt it using NodeJS [13]. 

The mobile application in this work serves as a companion app for PWM devices. This application can generate water tokens by accessing the water token vending server. It can also show the PWM’s remaining water balance and establish connections with PWM devices to add more water. Additionally, it provides users with a record of their token transactions. The decision to create a mobile application is to help customers interact with the PWM devices. It highlights the potential of mobile apps to enhance customer satisfaction by delivering personalized information tailored to meet consumer requirements [14]. This proposed solution also changes the purchasing method of the water payment and billing system. Figure 1 depicts the difference between the current system and the proposed one. 

With the current system (postpaid), PDAM sends its officers to record customer’s water usage at the end of each billing cycle. Subsequently, the officers will turn the water usage into water bills according to the customer’s tariff classification. Customers can pay their bills using various payment methods, including e-commerce, convenience stores, or the Quick Response Code Indonesian Standard (QRIS) [15,16,17]. If the customer refuses to make an on-time payment, a penalty will be imposed, and their bill will accumulate further if it remains unpaid during the next billing period.

With a prepaid system, PDAM does not need to send their officers to record water usage because the bills are paid upfront when the consumer purchases the water token. The PWMs will use a solenoid valve to shut off water access until the consumer redeems a water token acquired through the application. The proposed system can enhance transparency between users and PDAM by providing an upfront presentation of the token price and the quantity of water.

## 2. Materials and Methods

The Systematic Literature Review (SLR) methodology compiles published research from various online sources to provide a comprehensive overview of relevant publication subjects. By including research topics and keywords in search queries, a SLR maps and constructs a research strategy to find published publications, including those from journals and conferences [18]. 

The method has three phases, each with several tasks, carried out during the SLR process [19]. The initial phase entails planning to create search strings to query digital sources. The second phase, the review process, involves confirming their relevance by examining their titles and abstracts. After that, a full-text review or in-depth evaluation process was utilized to gather further information from the remaining publications relevant to our topic [19].

### 2.1. Systematic Literature Review (SLR)

A search string was created to explore Internet sources or repositories comprehensively [20]. The terms “prepaid water meter,” “flow meter,” and “water meter” were used, and the publication period was limited to the previous decade. This SLR aimed to investigate the various methodologies used in PWM development. Furthermore, the search terms were applied to Scopus and Google Scholar as online sources. Table 1 summarizes the results of the search string across these platforms.

The search yielded 435 publications from two repositories and utilized the Preferred Reporting Items for Systematic Reviews and Meta-Analyses (PRISMA) method. Figure 2 describes the PRISMA results for this SLR.

The PRISMA method identified 163 duplicate publications, as well as 272 non-duplicate publications. After that, 196 publications were excluded because they were unrelated to this work. Then, the remaining 272 publications were categorized as “include” or “exclude” to filter out publications that did not address the research topic [18]. The result is that 61 out of 76 publications were excluded due to reasons such as “wrong outcomes” (N = 44), “wrong study design” (N = 16), and “cannot download” (N = 1). This resulted in generating 15 relevant publications on the research topic, which will be utilized in this work.

These 15 publications will be used as references to highlight the main innovations in our research. They were assessed based on various factors, including flow sensors, meter types, communication mechanisms, standardization, and applications designed to aid research.

Table 2 summarizes the findings from all 15 publications included in the studycompared with our work that highlighted with green color. It shows that a wide variety of approaches and materials were used to reach similar results. 

There are several different ways to deal with sensor factors. Harry used Optical Character Recognition (OCR) with an ESP32-CAM sensor, Liu used a non-magnetic sensor, Akor added the VATS JT-121 paddle wheel flow sensor, and Yao made a system with an ultrasonic sensor and a TDC-GP30 chip. Still, this system does not use the prepaid mechanism [21,22,23,24]. In addition, other publications frequently employ hall-effect sensors such as G1/2, YF-S201, and YF-S403 [25,26,27].

There is a lack of uniqueness in the choice of meter type, as most publications use common varieties like mechanical and electronic meters. However, several publications employ different strategies in their communication methods, such as the utilization of InfraRed (IR) by Nurhayata et al. and the use of NFC by Chengyu et al. In contrast, several publications use more widely used communication methods, such as GSM/GPRS and LoRa [28,29]. Of the 15 publications, only three explicitly state the standards they use for transfer standardization. Nurhayata et al. utilize the Substitution Model Encryption technique to encrypt and decrypt a 12-digit token. The decryption process involves using the customer ID and water volume [28]. Chengyu et al.’s publications utilized Near-Field Communication (NFC) security standards to enable the secure transmission of data from mobile devices via NFC to the water meter. This is achieved by utilizing two NFC security standards: Shared Secret Services (SSE) and Secure Channel Service (SCS). SSE utilizes shared secrets to guarantee that peer NFCs have exclusive access, while SCS acts as a protective channel to secure communication in both directions over the channel [29,36]. 

Regarding companion applications, Liu’s use of the client terminal and Akor’s use of RFID are notable approaches to how users interact with the water meter device. These approaches stand out from other publications that commonly use websites or mobile applications to interact with the system. [22,29].

According to Table 2, the PWM in this work is unique because it employs two inductive-capacitive (LC) sensors to collect data from an electromechanical meter and transmit it over Bluetooth Low Energy (BLE) to a mobile application that acts as a companion app. These LC sensors will measure the rotation of a metal disc inside the water meter to keep the PWM device’s production cost affordable. 

Unlike the direct connection type used in other studies, this work uses two different communication channels: BLE and GSM. BLE will be utilized in field practice to establish a connection between a mobile application and the PWM. Moreover, mobile applications will utilize GSM to establish communication with a back-end server to process various application transactions, including user authentication, user transactions, and the generation of water tokens. 

This type of communication makes implementing the system quite low-cost, as it does not require external support equipment like gateways and communication towers. This work also utilizes the Standard Transfer Specification (STS) protocol to ensure standardized transfers when generating the 20-digit water token.

In summary, this work’s novelty lies in implementing a tokenization system and connecting it to the mobile application via BLE. The proposed system can also be implemented at a low cost, making it particularly suitable for use in Indonesia.

### 2.2. Mobile Application Development

L-Connect uses design thinking (DT) concepts to develop an innovative strategy for a mobile app. DT acts as an intermediary between human needs and technological possibilities, combining the innovation of a designer with an established methodology to create a human-centered design [37,38]. Implementing this user-centric strategy significantly changes how organizations handle application development and service delivery, ultimately leading to increased user satisfaction [39]. 

The design thinking process comprises five essential stages: empathizing, defining, ideating, prototyping, and testing [40]. On a small scale, the method includes recognizing the issue, setting precise criteria, brainstorming solutions, creating models, and performing thorough testing. The macro approach meets particular requirements by creating prototypes that solve the identified problem [41]. See Figure 3 for a visual depiction of these stages.

Based on Figure 3, each phase is explained as follows:Empathize is the first phase of DT. The development team utilizes Empathize to gather data from relevant stakeholders and determine the solutions users need, using empathy to identify problems [41];The Define phase analyzes the data gathered during the Empathize phase to determine user requirements [41];During the Ideate phase, the development team generates ideas through brainstorming to address the problems identified in the previous phase [41];The Prototype phase, also known as the experimental phase, involves creating a mobile application prototype. At this point, the mobile application and back-end development process begins [41];The final phase is Testing, which assesses the application prototype’s functionality and performance. [41].

## 3. Results

This section provides an in-depth overview of the Android application’s development process, employing the design thinking methodology. It begins with collecting user requirements and progresses through the prototyping phase. 

### 3.1. Empathize

During the empathize phase, surveys were conducted from various sources to understand potential general requirements and outcomes, which included observations and interviews. Sentiment analysis was utilized to analyze tweets containing terms like “PDAM”, “PDAM water”, and “meters”. As a result, 10,021 data points were collected. Twitter was chosen as the platform for analysis due to its popularity among PDAM consumers who share their ideas through written language. The observation results will serve as assessment data for PDAM, focusing specifically on customer feedback regarding PDAM meter usage.

After that, the collected data were examined, focusing on positive and negative attitudes, with neutral and ambiguous sentiments excluded. Figure 3 illustrates a word cloud reflecting negative sentiments.

Figure 4 reveals that the terms “water”, “PDAM”, “runout”, “payment”, and “billing” are associated with the five most common negative thoughts. These research findings indicate that water and payment issues are customers’ primary concerns. To explain and strengthen this work, the investigation began by examining bigrams.

Bigrams were employed to detect common negative sentiments in the dataset, as illustrated in Figure 4 [42]. Sentiments were categorized as “ PDAM Payment (*Pembayaran* PDAM)”, “Pipe Leakage (*Kebocoran pipa*)”, “Turbid Water (*Air Keruh*)”, and “Meter Records (*Pencatatan meter*).” Figure 4 illustrates how N-Grams categorize common words associated with negative sentiments.

According to Figure 5, the primary concern for customers is “PDAM Payment”, which is related to considerable differences in PDAM water bills compared to previous billing periods. In addition to surveys and observations, meetings were held with PDAM Tirta Pakuan Bogor City, yielding several findings including the following.

PDAM faces challenges in billing multiple consumers, prompting the need to develop a token-based prepaid system for water meters;Specifications for a prepaid token-based water meter system are being drafted.

All outcomes will be combined and analyzed from this data to identify the problems faced by PDAM and its customers.

### 3.2. Define

Based on the outcomes identified in the Empathize phase, a PWM system was suggested to solve this problem. Based on the results from the previous phase, functional and non-functional requirements for this application were developed. Functional requirements were utilized to create the front-end application, translating necessary application programming interface (API) requests.

Non-functional requirements were incorporated to improve the application’s functional features and ensure its smooth operation. Table 3 details the application’s functional and non-functional requirements.

### 3.3. Ideate

In Indonesia, there are two types of water meters, as shown in Figure 6: a conventional water meter on the left side and a water meter for AMR devices on the right. The standardized water meter for AMR-supported devices has been regulated by the Indonesian National Standard (SNI) number 2547:2019, which states there is a metal dial that becomes an attached complementary element with externally mounted sensors [43]. Based on this rule, every manufacturer in Indonesia can manufacture their own AMR-ready water meter that can be used with the PWM developed in this work [43].

Wireless inductive-capacitive (LC) sensors were employed on an electromechanical water meter to monitor water usage and accurately prevent water token misuse. These sensors detect the rotation of the metal disc in the water meter caused by water flow and enable precise monitoring of water usage [44,45]. Figure 6 shows a water meter with a metal disc.

The sensor operates independently and generates a magnetic field by oscillating at a given frequency with a pulse speed of 1 μs without requiring a physical connection to the metal disc. As the metal disc approaches and moves through the magnetic field, it absorbs magnetic energy from the inductor, reducing the oscillating signal, as depicted in Figure 7.

Figure 7 shows the arrangement of two LC sensors placed at 90-degree angles to each other, with the metal disc acting as the central point. This positioning is utilized to ascertain the metal disc’s motion through the sensors [45]. Figure 7 shows the variation in oscillation waves as the metal disc passes through the magnetic field [44].

This work uses two LC sensors to reduce errors caused by bounce-back effects when the solenoid valve shuts. The sensors are strategically placed to monitor the metal disc and the exposed side, guaranteeing precise measurements [46].

This PWM is powered by an internal lithium battery for all components, including the LC sensors, solenoid valves, and microcontrollers. The usage of internal batteries is based on the placement of the water meter, which is sometimes far from the electricity mains. The PWM employs a retrofit-style design, meaning it does not alter the existing water meter but rather enhances it to be compatible with prepaid functionality.

The mobile application serves as a companion app for the PWM devices. It has only one user role, which is the water customer. Users can log in or register a new account if they do not have one, generate water tokens, redeem water tokens using BLE, access information about the PWM devices, and view the history of generated water tokens.

The mobile application developed in this research is named L-Connect, derived from ‘Linflow Connect’, an abbreviation for the water meter product used in this study called Linflow. Table 4 details the application features generated in this research.

Figure 8 depicts the entire system architecture for this research. The mobile application was constructed using the React Native framework, facilitating the user’s generation of water tokens. It can be accessed on mobile devices such as phones or tablets that use the Android operating system and have BLE support. The PWM devices utilize the HM-10 BLE module to communicate with the mobile application via a BLE protocol.

The front end, which functions as a mobile application, establishes an API connection with the mobile application back-end (MABE). The API sends a JavaScript Object Notation (JSON) that includes the required water token value and a 20-digit water token, as shown in Figure 8.

The MABE serves as a method of verifying users’ identities, keeping a record of all user transactions, and acting as an intermediary for generating tokens using an API. This helps enhance security by hiding the Token Vending Server’s (TVS) IP address. The MABE consists of three elements: the TVS, the back-end server for the mobile application, and the database for storing all application data.

The TVS component was implemented using the C programming language, while the back-end component was implemented using the Laravel framework, and the database was built using MySQL. All these components are deployed through an online hosting provider.

The MABE analyzes the JSON data from the mobile application, conducts verification checks on the credentials, and adds a secret key for the TVS. The back-end server then sends the modified JSON to the TVS for further processing. After receiving the data, the TVS examines the credentials and secret key contained in the JSON from the back-end server. Assuming the credentials are accurate, the TVS will transmit the produced 20-digit token to the mobile application via the back-end server.

After the mobile app receives the token from the back-end server, users can redeem it by connecting to a PWM device using BLE to send the 20-digit token data. To generate tokens, the vending token server uses a standard process. This process encompasses interfaces and protocols that conform to three Standard Transfer Specifications (STS) generated by the International Electrotechnical Commission (IEC): IEC 62055-41, IEC 62055-51, and IEC 62055-52 [47,48,49]. Figure 9 illustrates the flow of the token system.

Figure 9 displays the token system flow and is divided into two data-flow interfaces: the Android to Token Carrier Interface and the Token Carrier to Meter Interface. Those interfaces use a two-layered protocol: The Application Layer (APDU) adheres to the IEC 62055-41 standard, while the Physical Layer (TCDU) conforms to the IEC 62055-5x series standard.

The token vending server (TVS) uses the Token Credit Transaction Command, which includes several parameters such as Meter ID, Supply Group Code (SGC), Tariff Index (TI), Key Revision Number (KRN), and Key Expired Number (KEN), to generate a 20-digit water token. Figure 10 illustrates the transaction command.

The MeterID in Figure 10 is a unique 11-digit code used to identify the water meter. A specific utility receives the SGC, a unique 6-digit code, to identify meter payments within the supply domain. The TI parameter is used to calculate the rates for each category. The vending key version for key decoding utilizes a KRN, which is a single-digit number from 1 to 9. The KEN functions as a digital certification expiration date, guaranteeing the key’s validity.

Users can communicate with the PWM using the mobile application L-Connect. This application allows users to generate tokens from the token server and then redeem them at the PWM devices. Figure 11 depicts L-Connect’s flowchart.

The application has the capability to display transaction records made by users. When connected to the PWM devices, it can also show the remaining water balance and provide information about water usage.

### 3.4. Prototyping

During this phase, the application development process moves to building the actual system, encompassing the L-Connect application’s front-end and back-end. This serves as a solution to the problem identified in the previous phase. Five features have been incorporated into the application, aligning with the feature specifications determined during the Ideate phase. These features include the home screen, water token generation, water token redemption, transaction history, and water usage reports.

#### 3.4.1. Home Screen

Figure 12 illustrates the redirection of users to the L-Connect home screen after logging in or registering a new account. Users can see the remaining tokens on the PWM devices while connected on this screen.

Users also have access to three features: Token, MyMeter, and Pengaduan (reporting). At the bottom of the screen, there is a navigation bar containing buttons to access transaction history and profiles.

#### 3.4.2. Generating a Water Token

One of the main features of the L-Connect mobile application is the ability to generate water tokens. This feature allows users to generate water tokens based on their preferred water value settings. The water value is measured in cubic meters (m^3^) to provide users with a better understanding. Figure 13 shows the screen flow for generating water tokens using the token value selection.

To generate a water token, users can access token generation through the Token menu on the home screen. They then select the Buy Token option, enter their PWM information, and choose the desired water token value. The system then redirects users to the payment screen. If the transaction is successful, the mobile application will send a request to generate the token, and it will display the generated token from the vending token server, as illustrated in Figure 14.

#### 3.4.3. Redeeming a Water Token

This is the L-Connect app’s second main feature. It encompasses two redeem flows. The first flow directly redeems water tokens after a user’s transaction is completed using the redeem button. This flow can also be accessed from the transaction history. The second flow aims to redeem water tokens obtained from the Point of Sale (POS) system as water vouchers.

After a transaction is complete, users can press the ‘Redeem Token’ button displayed on the transaction details screen to redeem the water token. The mobile application will then send water token data using BLE to the PWM. Figure 15 describes the screen for redeeming water tokens. Alternatively, if users wish to redeem previously purchased water tokens, they can access them from the transaction history. 

Users can also redeem their water vouchers generated by the POS system using the dedicated ‘Redeem Token’ menu inside the ‘Token’ menu from the home screen, as shown in Figure 16. The rest of the application flow remains the same as the first flow.

#### 3.4.4. Transaction History and Details

This feature allows users to see purchased water tokens. It displays the transaction data briefly, including the transaction date, payment amount, and water token amount. Compared to the transaction history feature, the transaction details feature provides users with more information. This includes details such as the invoice number, transaction date, payment method, paid amount, water token amount, and the water token itself, as shown in Figure 17.

Users can access this feature from the bottom navigation bar on the home screen. This feature allows users to view all their own transactions and later redeem purchased water tokens when connected to a PWM device.

#### 3.4.5. Water Usage Report

Users can access the water usage report feature when connected to the PWM. This feature displays all remaining token data recorded by the PWM and connected to the mobile application using BLE. If the PWM is not connected to the mobile app, the data will not be recorded. Figure 18 illustrates the water usage report feature.

From the home screen, users can access this feature by clicking the ‘See Details’ button, which will take them to the relevant section. Users can view their monthly usage reports, including day, date, and usage count. They can also select a desired month to view their water usage for a different period. 

## 4. Discussion

The PWM device and the mobile application (L-Connect) have been created to fulfill the system specifications stated by PDAM. In addition, the mobile application has been deployed on a third-party server and installed on Android devices. Figure 19 illustrates the advancements made and the outcomes achieved.

BLE connectivity allows the PWM system to establish a connection with Android phones. This connection enables the retrieval of tokens and data from the PWM device, such as battery status and remaining token information.

A comprehensive set of tests was conducted to ensure the system’s functionality before its public release. The tests spanned both the front-end and back-end components of the Android application and PWM device. The evaluations were placed at various stages of creation and subsequent testing using a wide range of testing instruments. 

Usability testing (UT) was utilized to assess the front-end, encompassing both the user interface (UI) and user experience (UX) components. Furthermore, a series of tests were conducted to evaluate the computational efficiency of the API back-end in the mobile application. The PWM device was also tested to ensure that the error rate from LC sensor readings complies with local laws.

### 4.1. Front-End Testing

During this step, usability testing (UT) was conducted on the user interface of the L-Connect application, with Maze used as a tool to facilitate the process. Maze employs the Maze Usability Score (MAUS), determined based on the average Mission Usability Score (MIUS). A total of 31 testers participated in this user testing phase, resulting in an MAUS score of 80 out of 100. 

The documentation published by Maze categorizes the MAUS score into three primary parts: low (MAUS < 50), medium (MAUS = 50–80), and high (MAUS = 80–100) [50]. According to this categorization, the usability score of 80 is categorized as “high” based on the MAUS score. Therefore, the mobile application’s UI and UX can be considered of good quality for end users.

### 4.2. Back-End Testing

The load testing of the mobile application back-end (MABE) was performed using Postman to assess the performance of the application programming interface (API) in producing water tokens for consumers during the purchase process. The primary objective of this load test was to quantify the API’s response time in milliseconds (ms) by utilizing the Postman Performance Tester from Collection Runner to continually call the API endpoint according to the test scenario.

The test scenario was set up with a fixed load profile consisting of five virtual users (VU) and lasted for 5 min. Considering the expected number of PDAM users in a district meter area (DMA), the testing was limited to around 500 requests to the API in total. Additional information about the test results is described in Figure 20.

Upon completion of the test, a cumulative count of 515 requests were transmitted to the back-end system, yielding a throughput rate of 1.68 requests per second. A mean response time of 1.973 milliseconds was recorded. 

Among the 515 requests made, a total of 45 replies were identified as errors, resulting in an error rate of 8.74% when expressed as a percentage. The 500 Internal Server Error was the main source of the issues.

### 4.3. LC Sensors Accuracy Testing

The PWM devices were measured with 100 L of water and repeated 30 times to assess the performance of the LC sensors. Throughout the testing process, we positioned the LC sensors of the PWM device, orienting them downward towards the metallic disc on the water meter. Figure 21 illustrates the arrangement of the LC sensors on the PWM device.

Over 30 test runs, the results show that the LC sensor readings are accurate between 0% and 2% error margin when the water temperature stays between 25 °C and 30 °C and the flow rate ranges from 480 to 500 L/h. Figure 22 describes the testing results for LC sensor reading accuracy.

According to Figure 22, the testing results show the error rates range from 0.0% to 2%, with an average error rate of 1.33%. The error rate of 1.33%, with a maximum error of 2%, is considered acceptable according to the SNI number 2547:2019 article 5.2. The 2% inaccuracy occurred due to the water meter itself, as the PWM reading depends on the recorded water usage from the water meter device [40].

The SNI number 2547:2019 article 5.2 standard specifies that the maximum allowable error is ±5% for flow rates between 32 and 88 L/h and ±2% for flow rates over 128 l/h [43]. The water meter utilized in this work is additionally certified by the SNI and has undergone testing by a laboratory accredited by the National Accreditation Committee (KAN) with code LP-579-IDN. Report number 08/LHU/MA/VIII/2023 and the same SNI number also support the meter’s certification.

Several publications have shown higher error rate of 2.87% and 3.54% when using flow meters equipped with a magnetic-based sensor known as the hall-effect sensor. This error rate was observed after measuring 1 L of water and taking 13 readings [25,26].

According to the test results, the PWM device meets the SNI standard, with an error of 2% and an average error of 1.33% [43]. This error falls below the SNI’s maximum permissible error and performs better than other hall-effect sensors.

### 4.4. Battery Usage Analysis

To minimize power usage and optimize cost-efficiency in developing the PWM device, the device employs an STM32L431CBT6 microcontroller, classified as a low-power microcontroller unit (MCU) [10,51]. Moreover, the PWM is powered by a lithium battery model LI-SOCL2 ER26500, which has a voltage of 3.6 V and a capacity of 8500 mAh. [8]. 

The battery lifespan was determined by measuring the battery usage with a power profiler and a multimeter to evaluate each state’s average current amount and duration. The measurements were conducted in three different states: during deep sleep with LC sensor readings; during BLE communications, which included pairing and data transmission; and during the activation of the solenoid valve.

Considering parameters such as the relationship between capacity and current, an operating temperature of 25 °C, and a cut-off voltage of 3.6 volts, it is possible to use only 80% of the initial capacity. This would leave a remaining capacity of 6800 mAh from 8500 mAh [52]. 

To ensure the PWM can operate for a minimum of 5 years, the battery usage period calculation was extended to 6 years for this test scenario to account for an amount of tolerance. This 5-year threshold aligns with the Indonesian local regional regulation of West Bandung Regency number 1 of 2018 regarding the implementation of retest and retribution for test/retest services which mandate that water meters must be reset and retested every 5 years. During each reset, the battery for the PWM was also replaced [53]. Therefore, Equation (1) can be used to determine the maximum current allowed in a single working cycle.
(1)$$I_{avg}\left(\max\right)=\frac{Usable\;Battery\;Capacity}{Time}\;I_{avg}\left(\max\right)=\frac{6800\;\mathrm{mAh}}{6\;\mathrm{years}\;\times \;365\;\mathrm{days}\;\times \;24\;h}=0.129\;\mathrm{mA}\;\mathrm{or}\;129\;\mu A$$

According to Equation (1), the mean current during a single operational cycle, which includes periods of deep sleep with LC sensor readings, BLE communications, and activation of the solenoid valve, should not surpass 129 uA for the PWM to have a lifespan of 6 years.

During a single operational cycle, the PWM remains in a low-power state (deep sleep) for a longer duration, with less frequent high-power activities (BLE communications and actuating the solenoid valve) to conserve energy. Equation (2) can be used to compute one operational cycle.
(2)$$I_{avg}(\max)=I_{sd}+\left(\frac{n\times Tds}{n\times Tds+Ttd+Ta}\times I_{ds}\right)+(\frac{n\times Ttd}{n\times Tds+Ttd+Ta}\times I_{td})+\;\left(\frac{Ta}{n\times Tds+Ttd+Ta}\times I_{a}\right)$$

Equations (1) and (2) have the following explanations:
$I_{avg}$(max) = maximum allowed average current (microamperes);$I_{sd}$ = battery self-discharge (microamperes);Tds = deep sleep duration (seconds);$I_{ds}$ = the average deep sleep current (microamperes);Ttd = data transmission duration (seconds);$I_{td}$ = the average current of data transmission (microamperes);Ta = duration of turning on the solenoid valve (seconds);$I_{a}$ = average current to turn on the solenoid valve (microampere);Tt = total duration of one cycle;n = the deep sleep state’s repetition value.

To get the lowest operational cycle necessary to achieve the maximum average current limit of 129 μA, the value of n can be calculated using Equation (2). On average, each state consumes a different current level with a different duration, as described in Table 5.

According to Table 5, the value of n can be determined by applying Equation (2). Equation (3) describes the complete calculation.
(3)$$129=10+\left(\frac{n\times 47.97}{n\times 47.97+1.5+13.2}\times 28.79\right)+(\frac{1.5}{n\times 47.97+1.5+13.2}\times 9100)+\;\left(\frac{13.2}{n\times 47.97+1.5+13.2}\times 30,290\right)=95.145$$

Using Equation (3), the calculated value of n is 95.145. Since n represents the number of deep sleep state repetitions, it is rounded to the nearest whole number, yielding n = 95. Consequently, the PWM must remain in the deep sleep state for 95 iterations, equivalent to approximately 75.95 min or 4557 s. Following this, BLE communications and solenoid valve activation can occur.

Extending the deep sleep state duration allows the PWM to achieve higher energy savings for the solenoid valve activation. Figure 23 illustrates the relationship between the duration of the deep sleep repetition condition and the actuator’s overall activation.

Figure 23 shows that if 95 deep sleep state repetitions occur in one operational cycle, the solenoid valve can actuate approximately 3900 times in 6 years, or around 50 actuations. This is because the longer the PWM device is in deep sleep, the less current it consumes, as shown in Figure 24.

With 95 repetitions or 75.95 min in the deep sleep state, the average current consumption is only 45 μA, far lower than the maximum allowed current (129 μA). With one year added as tolerance, customers can refill the water token balance an average of two times per month. The battery can last up to 6 years with minimal power usage at 129 μA.

With the PWM, the minimum amount of water to be purchased will be set at 10 m³. This aligns with the current billing policy that treats usage below 10 m³ uniformly. With water consumption at approximately 19,350 lpd, which is equal to 19.35 m³ per month, water customers will refill the PWM with a balance of 20 m³ per month or 10 m³ twice per month [5]. 

Assuming the purchase is made when the water balance in the PWM is depleted, the PWM will perform the open–close actuation twice per month. Given a capacity of 50 actuations per month, the PWM battery is projected to last for 6 years. Therefore, it can be concluded that the PWM battery will remain functional throughout the five years before PDAM requires a reset.

### 4.5. Cost–Volume Profit Analysis

The cost–volume profit analysis (CVPA) model assumes that variable costs and selling prices per unit remain constant. This assumption establishes a linear relationship between total revenue and expense that is independent of changes in volume.

The profit function, denoted as P(x), is defined as the function that represents the financial gain. The revenue function is denoted as R(x). Then, the cost function is denoted as C(x). The variable cost for each prepaid water meter is IDR 915,000, while the fixed cost is IDR 32,060,000. The fixed cost includes charges for equipment, server management, and manpower. Therefore, the overall cost can be calculated as follows: C(x) = 32060000 + 915000x(4)

As shown in Equation (4), the profit function will multiply the variable cost by x and then sum up the fixed cost. Each prepaid water meter costs IDR 1,296,000, including commissioning and software installation. Thus, the function R(x) can be precisely defined as:R(x) = 1296000x(5)

Equation (5) outlines the profit function that determines the revenue for this product. The profit function itself can be calculated using this function:P(x) = 381000x − 32060000(6)

Equation (6) defines the profit function used for profit calculation. Figure 25 visually depicts the CVPA model, showcasing all the specified functions and the break-even point. 

The graph has four lines depicted in distinct colors: the red line represents the total cost function C(x), the blue line represents the total revenue function R(x), the dashed green line represents the profit function P(x), and the dashed gray line indicates the break-even point.

Figure 25 shows that selling 84 units reaches the break-even point. At this point, total revenue equals total costs, resulting in neither profit nor loss. PDAM can save an additional IDR 6,984,000 by compensating meter recorders and administrative officers for processing the billing system monthly, as the prepaid system automates these tasks through the PWM. However, PDAM must still cover the fixed costs of the PWM system described earlier.

### 4.6. The Low-Cost and Low-Power Prototype

The device employs the STM32L431CBT6 microcontroller and is powered by a lithium battery to achieve a PWM device with low power consumption, as shown in Figure 24. This device has a lifespan of 6 years, which exceeds the minimum usage period required by Indonesian local regulations. The device also has a low production cost, as depicted in Figure 25. Furthermore, the device utilizes two simple LC sensors to collect input data, resulting in a high reading accuracy with an average of 1.33% error. It also utilizes BLE to communicate with the mobile application and can actuate the solenoid valve. Along with the low-power and low-cost characteristics, the device uses the Standard Transfer Specification (STS) to standardize the tokenization mechanism, which enhances security levels [47,48,49].

## 5. Conclusions

This study presents a solution to address the challenges posed by PDAM’s postpaid billing system in water utility bills by introducing a prepaid water meter (PWM) system. In the Emphasize phase, it was noted that the primary complaints of PDAM customers revolved around payment issues. However, PDAM faces challenges in efficiently invoicing numerous consumers due to limitations imposed by the postpaid system. This prompted the development of a PWM device and a mobile application for redeeming water tokens. The Define phase outlined functional and non-functional requirements for the application.

After the completion of the Ideate phase, the technological solution for the PWM system was developed. The task encompassed the development of a device that utilizes LC sensors and a metallic disc affixed to the water meter to measure water use. Implementing a tokenization scheme required consumers to acquire water tokens through the mobile application before obtaining water from PDAM. The solenoid valve of the PWM system will automatically close when the PWM device identifies that the customer’s water quota has been exhausted and no water token has been redeemed. This method halts the movement of water until a token is exchanged. The process of token production follows the STS specification, whereas the communication protocol utilized for token redemption between the PWM device and the Android application uses BLE.

Subsequently, coding for the mobile application continued, and the PWM system’s back-end systems were constructed during the Prototyping phase. The testing phase commenced after developing and deploying all requirements to the web-hosting platform. This phase entailed a comprehensive evaluation of all system components to evaluate their performance and compliance with the defined requirements. The findings from the testing phase indicated satisfactory operation of the LC sensors in accurately measuring water usage from conventional electromechanical water meters equipped with metal discs.

This study successfully developed a low-cost and low-power prepaid water meter (PWM) consisting of a PWM device, LC sensors, a mobile application called L-Connect, and a MABE. All these components were created to comply with the Standard Transfer Specification (STS). The evaluation of the PWM device demonstrated a notable degree of accuracy in measuring the LC sensor, exhibiting an error rate ranging from 0% to 2% with an average error of 1.33%. In contrast, the LC sensors were positioned vertically on the PWM device, looking downwards towards the metal disc of an electromechanical water meter. The L-Connect application, developed utilizing the design thinking technique, received a score of 80, indicating a ‘High’ classification based on MAUS metrics. The MABE was subjected to a comprehensive stress testing procedure using Postman Runner, which spanned 5 min and involved five virtual users. The outcome of this testing was an average request rate of 1.68 requests per second, resulting in a total of 515 requests sent to the MABE. However, 45 error replies were recorded, representing 8.74% of the data. These responses were mainly attributed to 500 Internal Server problems.

To guarantee the durability of the PWM device for five years, the battery analysis was extended to 6 years, allowing for a 20% margin to ensure customer satisfaction within the specified time range. The analysis shows that the PWM device can operate for six years using a work cycle that includes 95 occurrences of extended periods of inactivity, followed by transferring the BLE data and activating the solenoid valve.

In addition to the technical advantages, this work highlights the novelty of integrating a tokenization system with a mobile application via BLE. The proposed system offers a low-cost implementation, particularly suitable for Indonesia, using a two-channel communication system consisting of BLE and GSM. With these two-channel communications, the expenses associated with its implementation can be reduced since it does not require supporting equipment like gateways and communication towers. However, the proposed system also has the disadvantage that PWM data sent to the back-end server are not real-time. This is because the PWM must be connected to the mobile application via BLE before the application can send the received PWM data to the back-end server.

## 6. Patents

The research findings are protected by patents granted by the Republic of Indonesia Ministry of Law and Human Rights, with request number EC002023104582 and record number 000537537. The patents are valid until 2073, as stated in Article 72 of Law Number 28 of 2014 on Copyright. The lifespan is 50 years from the initial notice on 23 October 2023.

## Figures and Tables

**Figure 1 sensors-24-06762-f001:**
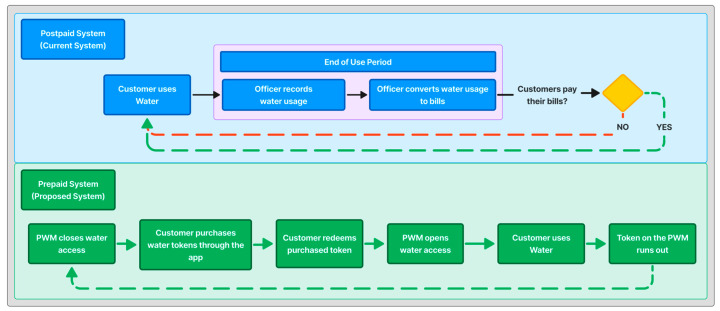
Water purchase flow comparison, with blue background represented current system and green background represented proposed system.

**Figure 2 sensors-24-06762-f002:**
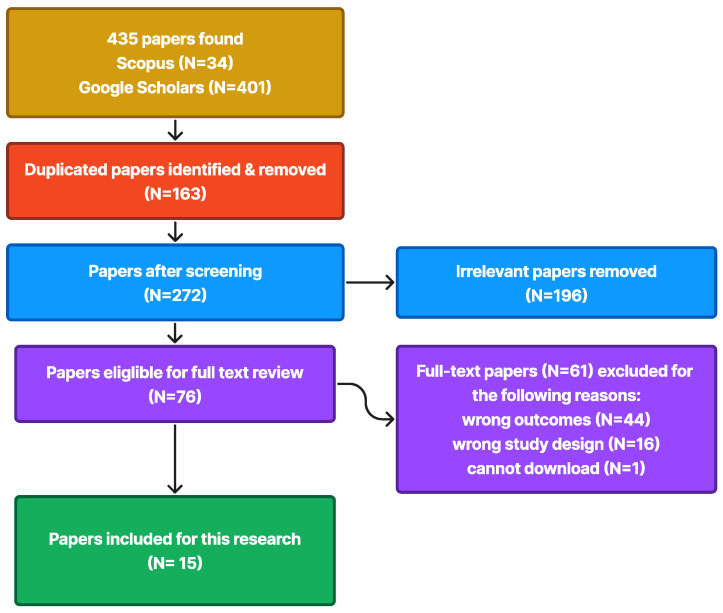
The publication filtration process in the SLR procedure is depicted in the PRISMA flowchart.

**Figure 3 sensors-24-06762-f003:**
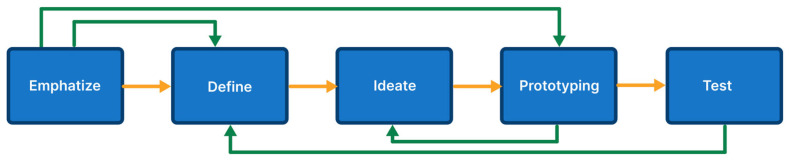
Design thinking phase.

**Figure 4 sensors-24-06762-f004:**
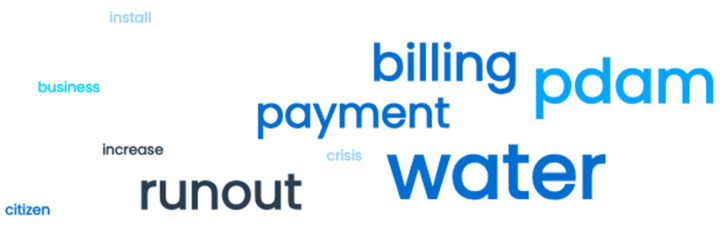
Word cloud of negative sentiments.

**Figure 5 sensors-24-06762-f005:**
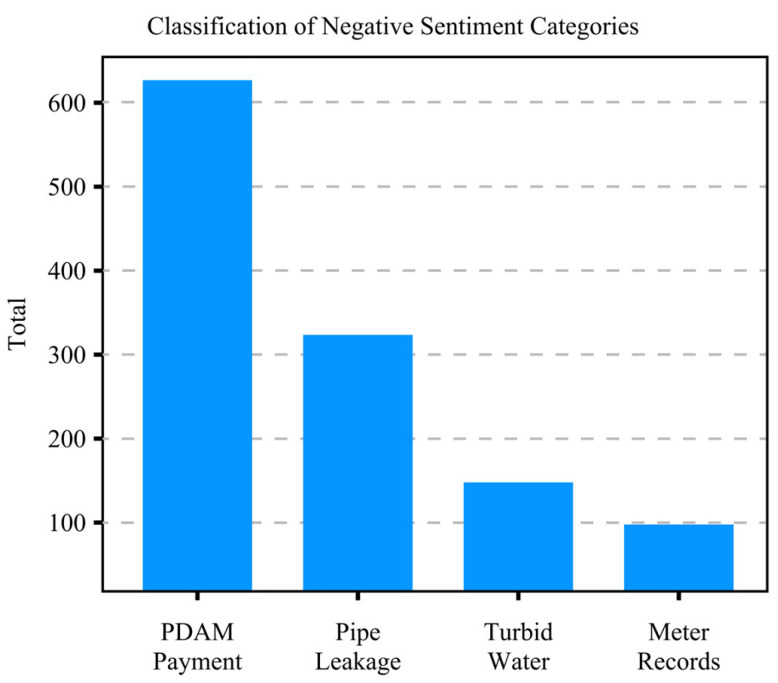
N-Grams visualization of negative sentiments.

**Figure 6 sensors-24-06762-f006:**
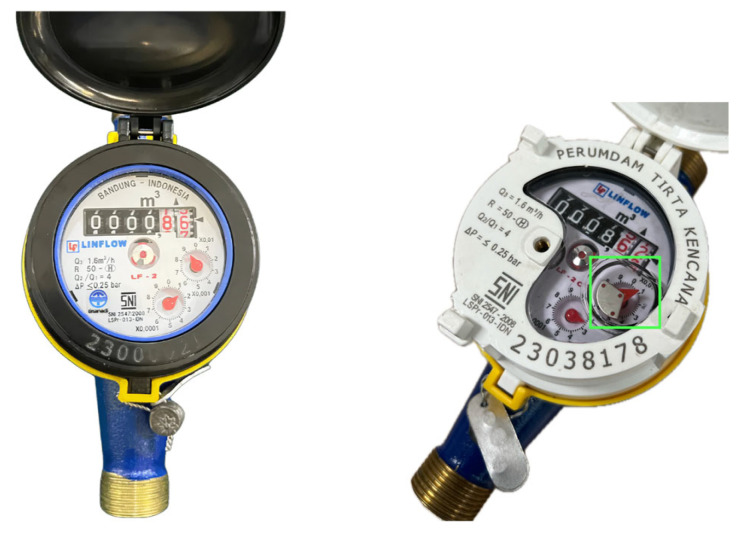
A conventional water meter (**left**) and a water meter with a metal disc designed for AMR readings with LC sensors (**right**).

**Figure 7 sensors-24-06762-f007:**
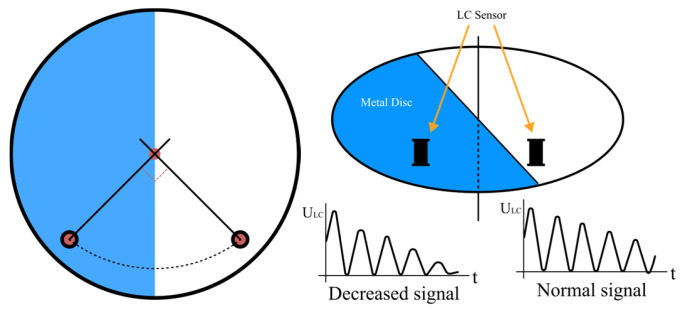
LC sensor placement as viewed from above (**left**) and the working principle (**right**).

**Figure 8 sensors-24-06762-f008:**
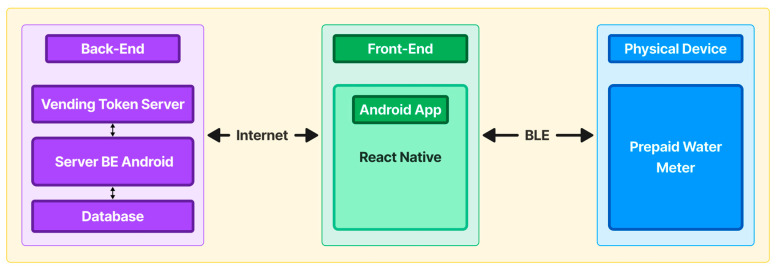
Mobile application system architecture.

**Figure 9 sensors-24-06762-f009:**
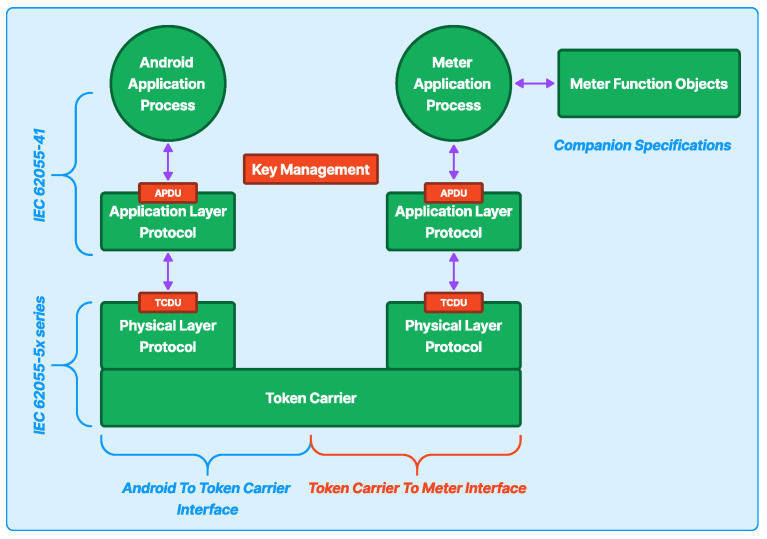
Token system flow.

**Figure 10 sensors-24-06762-f010:**
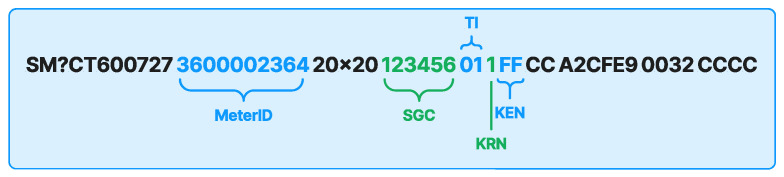
Token credit transaction command.

**Figure 11 sensors-24-06762-f011:**
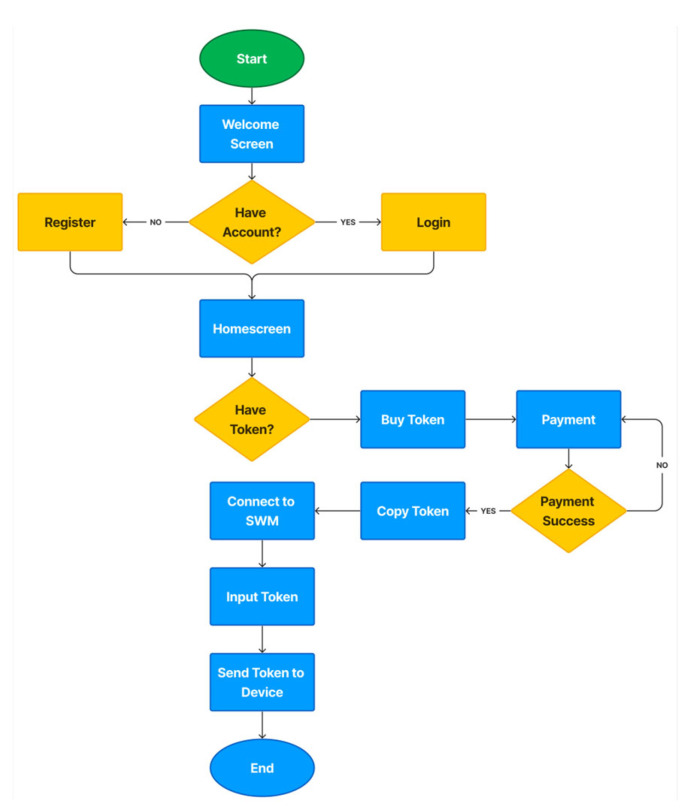
L-Connect flowchart.

**Figure 12 sensors-24-06762-f012:**
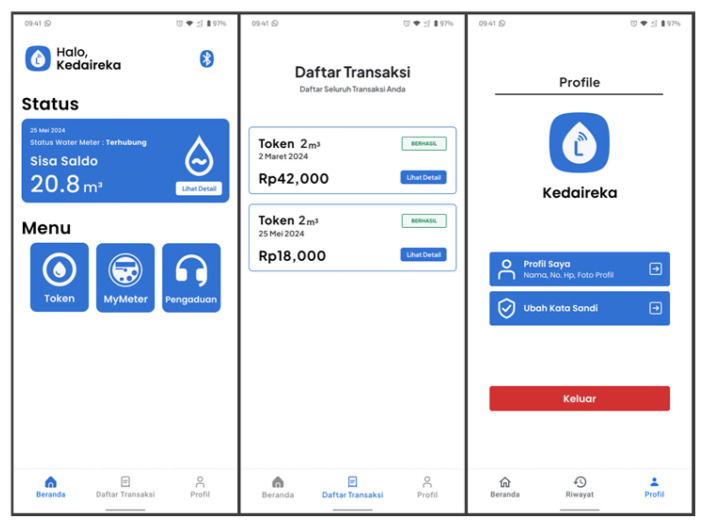
L-Connect home screen.

**Figure 13 sensors-24-06762-f013:**
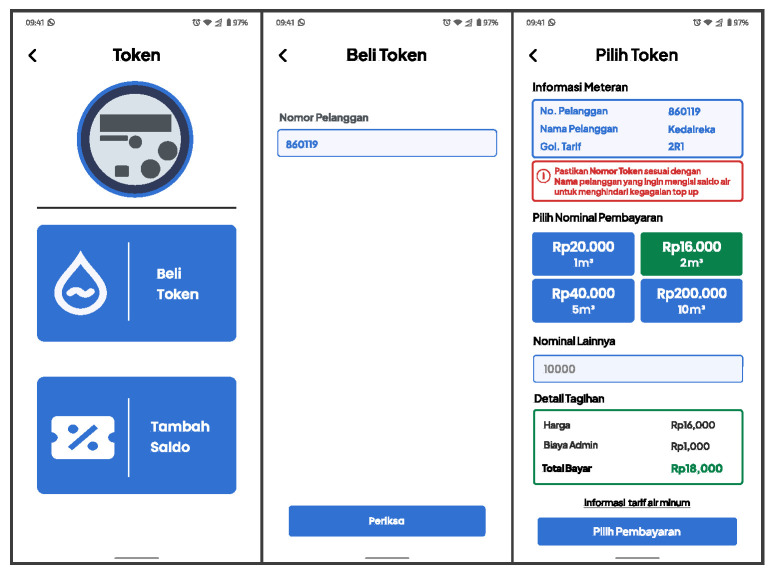
Water token selection screen.

**Figure 14 sensors-24-06762-f014:**
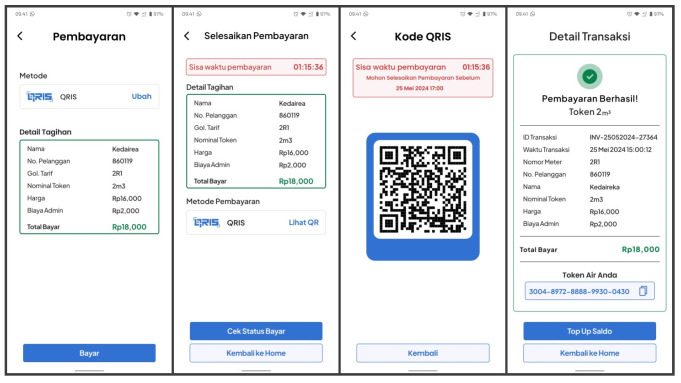
Payment process and the generated water token screen.

**Figure 15 sensors-24-06762-f015:**
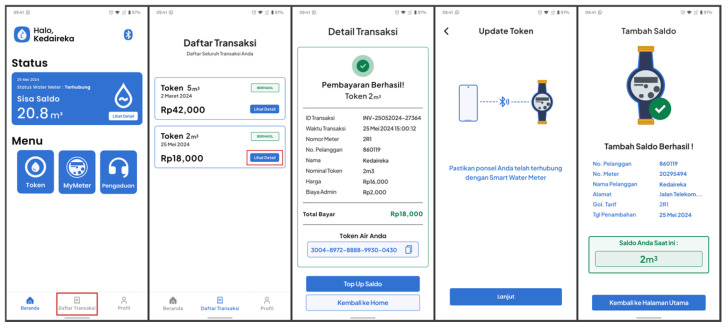
Redeeming the purchased token from the application.

**Figure 16 sensors-24-06762-f016:**
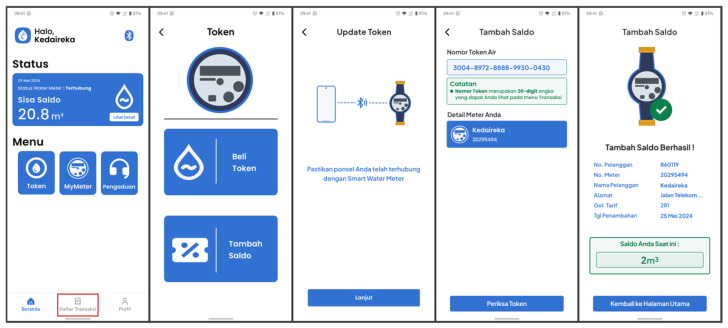
Redeem the purchased water token from the POS system.

**Figure 17 sensors-24-06762-f017:**
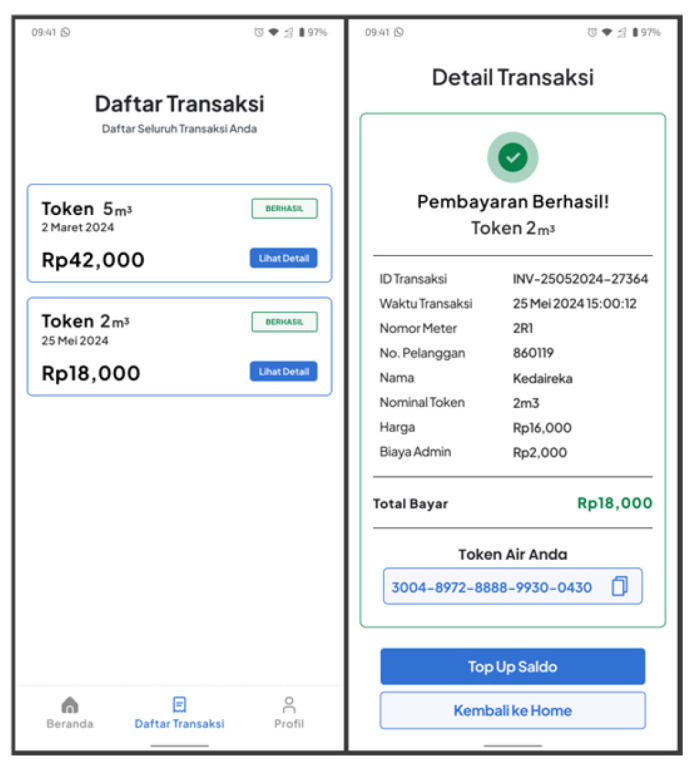
Transaction history and details screen.

**Figure 18 sensors-24-06762-f018:**
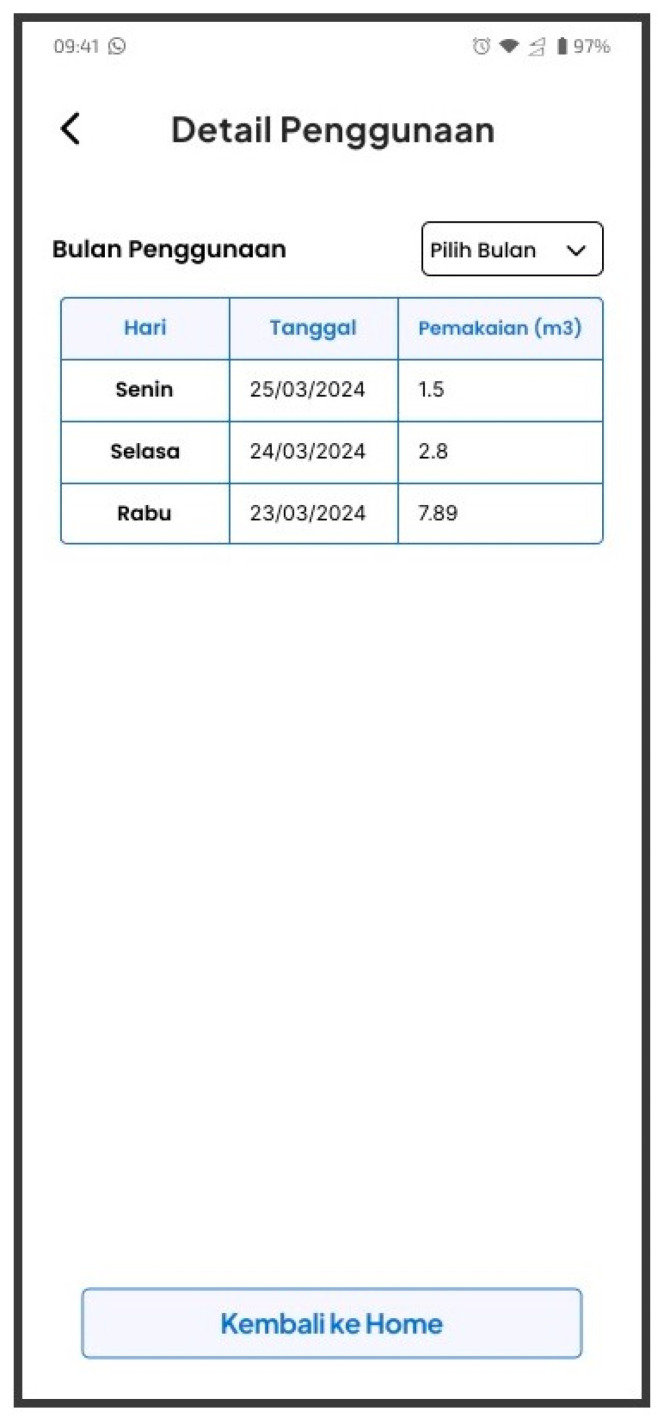
Water Usage Report.

**Figure 19 sensors-24-06762-f019:**
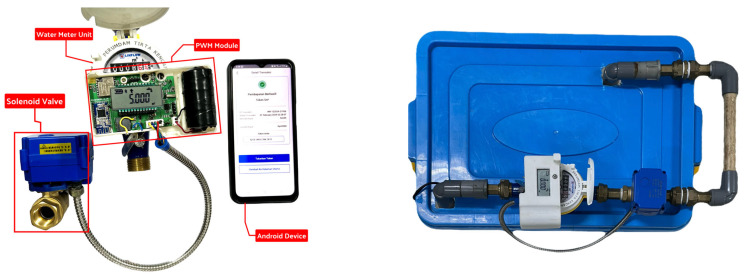
Android phones successfully generate 20-digit tokens with a PWM device (**left**) and a PWM device installed on a display device (**right**).

**Figure 20 sensors-24-06762-f020:**
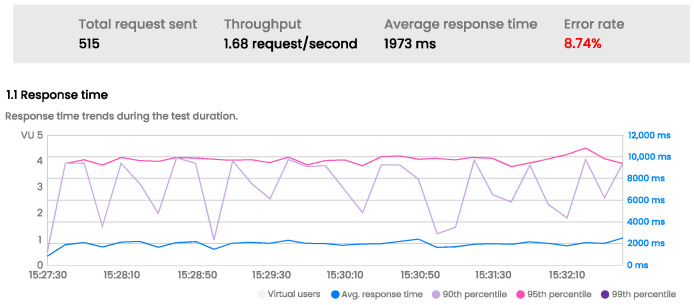
Postman API response time load test results in summary (**top**) and API performance graph (**bottom**).

**Figure 21 sensors-24-06762-f021:**
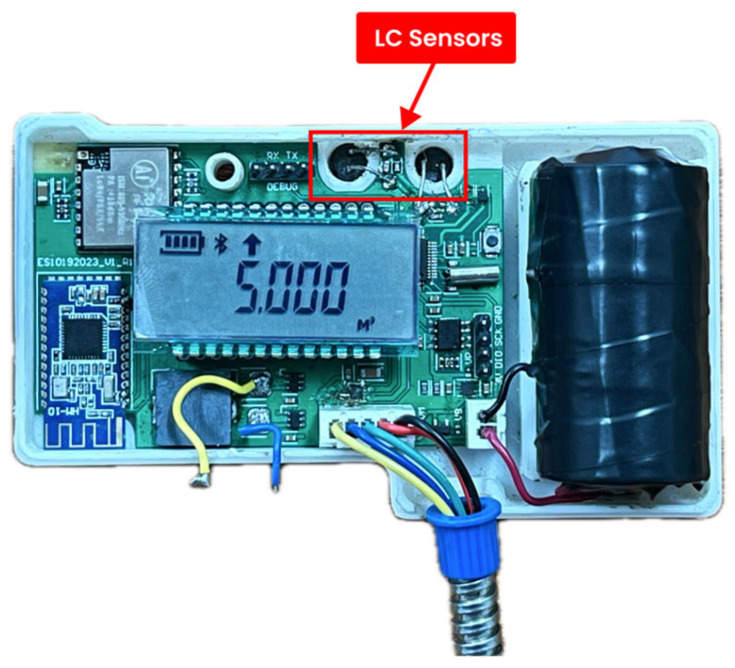
The LC Sensor placement on the PWM device.

**Figure 22 sensors-24-06762-f022:**
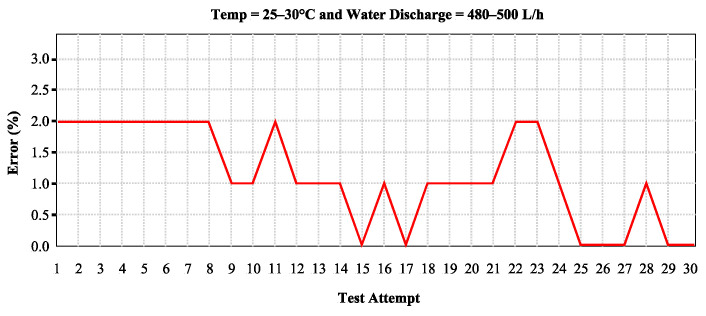
The PWM LC sensors accuracy testing.

**Figure 23 sensors-24-06762-f023:**
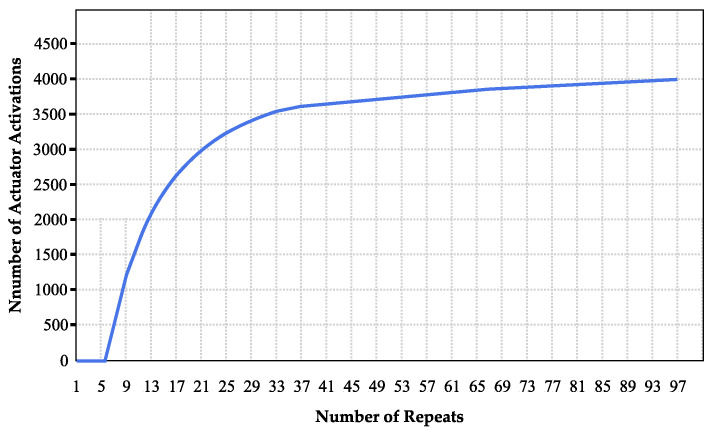
Relations between actuator activation and deep sleep repetitions.

**Figure 24 sensors-24-06762-f024:**
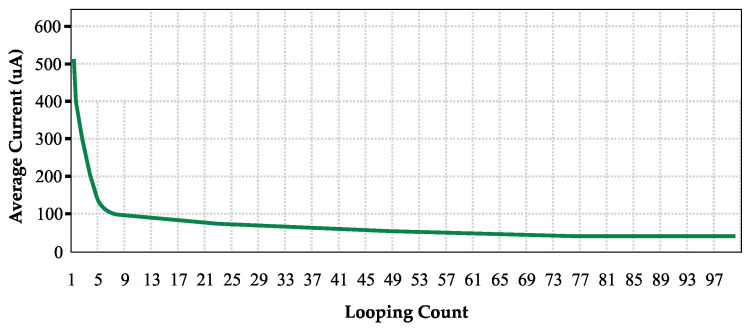
Relationships between used current (average) in μA and deep sleep repetitions.

**Figure 25 sensors-24-06762-f025:**
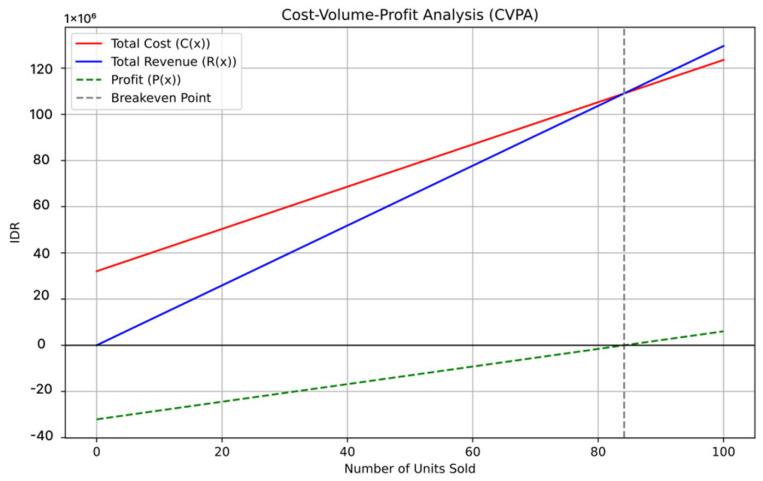
CVPA graph.

**Table 1 sensors-24-06762-t001:** Scopus and Google Scholars search results.

Database	Search String	Results
Scopus	flow meter AND water meter AND prepaid meter AND prepaid water meter	4
Scopus	prepaid water meter	30
Google Scholar	prepaid water meter	200
Google Scholar	flow meter AND water meter AND prepaid meter AND prepaid water meter	201

**Table 2 sensors-24-06762-t002:** SLR publications evaluation result.

Reference	Flow Sensor	Meter Type	Communication Method	Transfer Standardization	Application
[21]	ESP32-CAM	Mechanical	LoRaWan	-	Website
[22]	Non-Magnetic Sensor	Mechanical	NB-IoT	-	Client Terminal
[23]	VATS JT-121	Mechanical	GSM	-	RFID
[24]	Ultrasonic	Electronic	IR	-	-
[25]	G1/2	Electronic	GSM/GPRS	-	Mobile via Thingspeak/Web Arduino
[26]	YF-S201	Electronic	ZigBee & Wi-Fi	-	Website
[27]	YF-S403	Electronic	LoRa	-	Website & Mobile
[28]	G 1/2	Electronic	IR	Substitution Model Encryption	-
[29]	Flow Sensor	Mechanical	NFC	NFC Security SSE & SCH	Android
[30]	-	Electronic	GPRS	-	Website & Mobile
[31]	-	Electronic	GSM	-	USSD & Web Dashboard
[32]	YF-S201	Electronic	GSM/GPRS	-	Website
[33]	Hall Effect Sensor	Electronic	GSM/GPRS	-	Web Dashboard
[34]	Flow Sensor	Digital	Wi-Fi	-	Website & Mobile
[35]	Flow Sensor	Electronic	Wi-Fi	-	Website
This Work	LC Sensors	Electro Mechanical	BLE + GSM	Standard Transfer Specification	Mobile App

**Table 3 sensors-24-06762-t003:** Functional and non-functional requirements for the application.

Application Requirements	
**No**	**Functional Requirements**
1	System can show authentication to authenticate users.
2	System can show reading result from water reading unit.
3	System can show detailed data from water reading.
4	System can show an account management menu for customers.
5	System can show account registration for customers.
6	System can show a payment menu for customers to pay for water billing.
7	System can show a transaction history to show historical payments for customers
8	System can generate a token by vending the token server
9	System can connect to prepaid water meter devices using BLE to redeem tokens
**Non-Functional Requirements**	
1	System must use encryption to protect sensitive information.
2	System must respond quickly to user actions.
3	System should be available and fully functional 24/7 with minimal downtime.
4	System should be able to handle an enormous growing number of users and transactions without significant performance degradation.
5	System must have easy to use and intuitive user interface.

**Table 4 sensors-24-06762-t004:** Feature requirement for L-Connect.

No	Role	Priority	Feature
1	User	Medium	Login and registration
2	User	High	Generating water tokens
3	User	High	Redeeming water tokens
4	User	High	Transaction history and details
5	User	High	Prepaid water meter device information

**Table 5 sensors-24-06762-t005:** The PWM power consumption for each state.

State	Time	Avg. Current (μA)
Deep sleep	47.97 s	28.79
BLE communications	1.50 s	9100
Actuate solenoid valve	13.20 s	30,290

## Data Availability

Data presented in this study are available on request from the corre- sponding author. The data are not publicly available due to internal policies of the Institution.

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
