# Peer review of "Mobile Application Development for Prepaid Water Meter Based on LC Sensor"

_sensors, 2024, doi:10.3390/s24206762_

Round 1
Reviewer 1 Report (Previous Reviewer 1)
Comments and Suggestions for Authors
The authors have revised the manuscript according to the comments. This paper can be accepted.
Comments on the Quality of English LanguageMinor editing of English language required.
Author Response
Response to Reviewer 1 Comments |
||
1. Summary |
|
|
Thank you very much for taking the time to review this manuscript. Please find the detailed responses below and the corresponding revisions and corrections in the re-submitted files. |
||
2. Questions for General Evaluation |
Reviewer’s Evaluation |
Response and Revisions |
Does the introduction provide sufficient background and include all relevant references? |
Can be improved |
We have addressed this feedback, and the revisions can be found on point-by-point response. |
Is the research design appropriate? |
Can be improved |
We have addressed this feedback, and the revisions can be found on point-by-point response. |
Are the methods adequately described? |
Can be improved |
We have addressed this feedback, and the revisions can be found on point-by-point response. |
Are the results clearly presented? |
Can be improved |
We have addressed this feedback, and the revisions can be found on point-by-point response. |
Are the conclusions supported by the results? |
Can be improved |
We have addressed this feedback, and the revisions can be found on point-by-point response. |
3. Point-by-point response to Comments and Suggestions for Authors |
||
Comments 1: The authors have revised the manuscript according to the comments. This paper can be accepted.
|
||
Response 1: Thank you for your positive feedback and for acknowledging our revisions. We greatly appreciate your time and effort in reviewing our manuscript. We are pleased to hear that the revised version meets your expectations. |
||
4. Response to Comments on the Quality of English Language |
||
Point 1: Minor editing of English language required. |
||
Response 1: Thank you for your feedback regarding the clarity of the English in our manuscript. Therefore we have use Grammarly Business with overall score 99 as depict on Figure 1 to ensure english language in our manuscript.
Figure 1 |
||
5. Additional clarifications |
||
We have addressed all feedback from Reviewer 1, as detailed in the point-by-point comments and suggestion above. |
Reviewer 2 Report (Previous Reviewer 2)
Comments and Suggestions for Authors
This manuscript (sensors-3221146) is a resubmission of the manuscript (sensors-3063451). The current version has been improved concerning the original. The authors have responded to all comments from the previous submission and review.
The authors of the manuscript set themselves the goal of developing a mobile application connected to a server to use tokenization for paying water bills in a prepaid manner. The developed solution aims to minimize social problems and reduce the costs of billing reading.
The authors of the manuscript used LC sensors for automatic reading of water meters and communication between the mobile application and the PWM device that controls the valve and reads the water meter readings.
The main part of the work consists in presenting the method of analyzing scientific articles containing keywords selected by the authors. To the reviewer's knowledge, preapid water meter systems come in many configurations. The use of LC sensors is also not innovative. Remote reading systems for water meters from leading manufacturers are based on LC sensors and magnetic induction (ITRON, Madallena, APATOR) or the Hall effect (Diehl Metering).
The idea of ​​prepaid meters is not new. There are many such solutions worldwide, both for the water sector and others. The novelty is the use of tokenization and connection with a mobile application. However, the authors presented a new solution adapted to Indonesian conditions.
I accept in present form.
Author Response
For research article
Response to Reviewer 2 Comments
|
||
1. Summary |
|
|
Thank you very much for taking the time to review this manuscript. Please find the detailed responses below and the corresponding revisions and corrections in the re-submitted files.
|
||
2. Questions for General Evaluation |
Reviewer’s Evaluation |
Response and Revisions |
Does the introduction provide sufficient background and include all relevant references? |
Yes |
The Reviewer 2 has confirmed that introduction provide sufficient background and include all relevant references. |
Is the research design appropriate? |
Yes |
The Reviewer 2 has confirmed that the research design appropriate. |
Are the methods adequately described? |
Yes |
The Reviewer 2 has confirmed that the methods adequately described. |
Are the results clearly presented? |
Can be improved |
We have addressed this feedback, and the revisions can be found on point-by-point response. |
Are the conclusions supported by the results? |
Yes |
The Reviewer 2 has confirmed that the conclusions supported by the results. |
3. Point-by-point response to Comments and Suggestions for Authors |
||
Comments 1: This manuscript (sensors-3221146) is a resubmission of the manuscript (sensors-3063451). The current version has been improved concerning the original. The authors have responded to all comments from the previous submission and review.
The authors of the manuscript set themselves the goal of developing a mobile application connected to a server to use tokenization for paying water bills in a prepaid manner. The developed solution aims to minimize social problems and reduce the costs of billing reading.
The authors of the manuscript used LC sensors for automatic reading of water meters and communication between the mobile application and the PWM device that controls the valve and reads the water meter readings.
The main part of the work consists in presenting the method of analyzing scientific articles containing keywords selected by the authors. To the reviewer's knowledge, preapid water meter systems come in many configurations. The use of LC sensors is also not innovative. Remote reading systems for water meters from leading manufacturers are based on LC sensors and magnetic induction (ITRON, Madallena, APATOR) or the Hall effect (Diehl Metering).
The idea of ​​prepaid meters is not new. There are many such solutions worldwide, both for the water sector and others. The novelty is the use of tokenization and connection with a mobile application. However, the authors presented a new solution adapted to Indonesian conditions.
I accept in present form.
|
||
Response 1: Thank you very much for your thorough review and for acknowledging the improvements made in this resubmission. We are pleased that the current version addresses the concerns raised in the previous round. We appreciate your recognition of the novelty of our work, particularly in adapting the prepaid water meter system with tokenization and mobile application integration to Indonesian conditions. Additionally, we have revised the manuscript to better highlight this novelty, especially the integration of the token system and its connection to the mobile application. These revisions can be found in the Systematic Literature Review (SLR) section on page 6, lines 185-187, the Conclusion section on page 22, lines 686-694, and the Abstract on page 1, lines 11-13. |
||
4. Response to Comments on the Quality of English Language |
||
Point 1: - |
||
Response 1: Reviewer 2 mentioned that they are not qualified to assess the quality of English in this paper. To ensure the manuscript meets the necessary language standards, we have thoroughly reviewed and revised the text for clarity and correctness. |
||
5. Additional clarifications |
||
We have addressed all feedback from Reviewer 2, as detailed in the point-by-point comments and suggestion above. |
This manuscript is a resubmission of an earlier submission. The following is a list of the peer review reports and author responses from that submission.
Round 1
Reviewer 1 Report
Comments and Suggestions for Authors
The authors did not revise the manuscript according to the comments. This paper can not be accepted.
Comments on the Quality of English LanguageEnglish very difficult to understand/incomprehensible.
Author Response
For research article
Response to Reviewer 1 Comments
|
||
1. Summary |
|
|
Thank you very much for taking the time to review this manuscript. I sincerely apologize for the oversight in the previous round, where I mistakenly did not attach the reply to reviewer files directly to the Round 1 submission (Manuscript ID: sensors-2956666). Instead, I included the reply files in the resubmission form when submitting the current manuscript (Manuscript ID: sensors-3063451). The revisions and suggestions listed below pertain to the current round. The revisions and suggestions from the earlier round have been included at the end of this document. However, I want to assure you that the current submission is the latest version of the manuscript, fully revised according to the reviewers' feedback and suggestions from Round 2.
Please note that all page and line references mentioned in this document, for both the earlier and current rounds of revisions, are based on the latest version of the manuscript. Please find the detailed responses below and the corresponding revisions in the resubmitted manuscript files. |
||
2. Questions for General Evaluation |
Reviewer’s Evaluation |
Response and Revisions |
Does the introduction provide sufficient background and include all relevant references? |
Not applicable |
Thank you for your evaluation. We have reviewed the introduction to ensure it provides sufficient background and includes all relevant references. If there are specific areas you believe need further revision, we would appreciate your guidance to make the necessary adjustments. |
Is the research design appropriate? |
Not applicable |
We believe the research design is appropriate for addressing the objectives of this study. However, we are open to further suggestions for improvement and welcome any specific feedback you might have. |
Are the methods adequately described? |
Not applicable |
The methods section has been carefully reviewed. If there are particular aspects of the methodology that require further elaboration, we would greatly value your detailed input to enhance the clarity and comprehensiveness of this section. |
Are the results clearly presented? |
Not applicable |
We have worked to present the results as clearly as possible. Should there be any areas that remain unclear or require additional explanation, we are fully prepared to address them in a revised submission. |
Are the conclusions supported by the results? |
Not applicable |
We have ensured that the conclusions are supported by the results. However, if you have identified any discrepancies or areas for improvement, we would appreciate your specific recommendations. |
3. Point-by-point response to Comments and Suggestions for Authors |
||
Comments 1: The authors did not revise the manuscript according to the comments. This paper can not be accepted. |
||
Response 1: Thank you for your feedback. I sincerely apologize for the confusion and any inconvenience caused. During the resubmission process, I mistakenly did not attach the reply to reviewer files directly to the Round 1 submission (Manuscript ID: sensors-2956666). Instead, the reply files were included in the resubmission form when submitting the current manuscript (Manuscript ID: sensors-3063451). As a result, it may have appeared that the manuscript was not revised according to the comments provided in the previous round.
I want to assure you that the current submission (Manuscript ID: sensors-3063451) is the latest version of the manuscript, which has been fully revised according to the reviewers' feedback and suggestions from Round 2. If there are any specific areas that you believe were not adequately revised, aside from the quality of English, we would greatly appreciate your guidance. We are committed to making further revisions as needed to meet the required standards. Thank you for your understanding, and We look forward to your feedback. |
||
4. Response to Comments on the Quality of English Language |
||
Point 1: English very difficult to understand/incomprehensible. |
||
Response 1: Thank you for your feedback regarding the clarity of the English in our manuscript. We sincerely apologize for any confusion caused by the language in the original submission. To address this issue, we have thoroughly revised the manuscript to improve the clarity and readability of the text. We hope that these revisions have made the content more accessible and easier to understand. If there are specific sections that remain unclear, we would appreciate your guidance so we can make further improvements. |
||
5. Additional clarifications |
||
We have addressed all feedback from Reviewer 1, as detailed in the point-by-point comments and suggestion above. |
Previous Round Response
Response to Reviewer 1 Comments
|
||
1. Summary |
|
|
Thank you very much for taking the time to review this manuscript. I sincerely apologize for the oversight in the previous round, where I mistakenly did not attach the reply to reviewer files directly to the Round 1 submission (Manuscript ID: sensors-2956666). Instead, I included the reply files in the resubmission form when submitting the current manuscript (Manuscript ID: sensors-3063451). The revisions and suggestions listed below pertain to the current round. The revisions and suggestions from the earlier round have been included at the end of this document. However, I want to assure you that the current submission is the latest version of the manuscript, fully revised according to the reviewers' feedback and suggestions from Round 2. Please find the detailed responses below and the corresponding revisions in the resubmitted manuscript files. |
||
2. Questions for General Evaluation |
Reviewer’s Evaluation |
Response and Revisions |
Does the introduction provide sufficient background and include all relevant references? |
Yes |
The Reviewer 1 has confirmed that the introduction provides sufficient background and includes all relevant references. |
Is the research design appropriate? |
Can be improved |
We have addressed this feedback, and the revisions can be found on page 4, lines 173-188. |
Are the methods adequately described? |
Yes |
The Reviewer 1 has confirmed that the methods adequately described |
Are the results clearly presented? |
Yes |
The Reviewer 1 has confirmed that the results clearly presented |
Are the conclusions supported by the results? |
Can be improved |
We have addressed this feedback, and the revisions can be found on page 20, lines 673-675 and page 22, lines 688-692. |
3. Point-by-point response to Comments and Suggestions for Authors |
||
Comments 1: Nice paper including literature review, the authors should consider ultrasonic sensors from Kamstrup A/S they are very accurate for water measuring including wireless mesh for data transmission. The overall architecture is good, perhaps they should do some cost benefit analysis for the entire proposes systems to see feasibility on terms of operational cost. |
||
Response 1: Thank you for pointing this out. We agree with this comment. Therefore, we have revised the manuscript to include ultrasonic sensor that can be found at page 4, lines 149-151. We also add cost benefit analysis or Cost Volume Profit Analysis (CVPA) that include all cost used in the proposed system and calculate breakeven point. The CVPA also included graph to describe it visually. The CVPA itself and can be found at section 4.5, page 20 line 607, to page 21, line 534. |
||
4. Response to Comments on the Quality of English Language |
||
Point 1: propose to use a reviewer for double checking correct English |
||
Response 1: Thank you for your feedback regarding the clarity of the English in our manuscript. We sincerely apologize for any confusion caused by the language in the original submission. To address this issue, we have thoroughly revised the manuscript to improve the clarity and readability of the text. We hope that these revisions have made the content more accessible and easier to understand. If there are specific sections that remain unclear, we would appreciate your guidance so we can make further improvements. |
||
5. Additional clarifications |
||
We have addressed all feedback from Reviewer 1, as detailed in the point-by-point comments and suggestion above. |
Response to Reviewer 2 Comments
|
||
1. Summary |
|
|
Thank you very much for taking the time to review this manuscript. Please find the detailed responses below and the corresponding revisions and corrections in the re-submitted files.
|
||
2. Questions for General Evaluation |
Reviewer’s Evaluation |
Response and Revisions |
Does the introduction provide sufficient background and include all relevant references? |
Must be improved |
We have addressed this feedback, and the revisions can be found on page 2, lines 63-64. |
Is the research design appropriate? |
Must be improved |
We have addressed this feedback, and the revisions can be found on page 5, lines 174-178. |
Are the methods adequately described? |
Must be improved |
We have addressed this feedback, and the revisions can be found on page 6, line 204. |
Are the results clearly presented? |
Must be improved |
We have addressed this feedback, and the revisions can be found on page 17-21, section 4.4, 4.5, and 4.6. |
Are the conclusions supported by the results? |
Must be improved |
We have addressed this feedback, and the revisions can be found on page 21, line 673-674. |
3. Point-by-point response to Comments and Suggestions for Authors |
||
Comments 1: The technical parts about the sensor and the test of it could only occupy two pages at most. The sensor used was a simple application of a mature device as well. Thus, the novelty of the paper and its fitness to the journal of Sensors are very questionable. |
||
Response 1: Thank you for pointing this out. We agree with this comment. Therefore, We have made the technical and the test part of sensor used in this manuscript shorter. We also add explanation about why simple sensor was used in this manuscript and provide further explanation about novelty of the paper that located at Page 5, lines 174-188.
|
||
Comments 2: Sections 2.2, 2.3, 2.4, the other 2.2 (should have been 2.5), and 3.1 are unnecessary and have made the manuscript read like some thesis instead of a journal paper. Table 2 itself is sufficient to introduce the background. |
||
Response 2: Thank you for pointing this out. We partially agree with this comment and changed our work to emphasize this comment. We changed sections 2.2, 2.3, 2.4, and 2.5 into only two sections consist of sections 2.1 and 2.2. This changed can be found at page 3, line 119 for section 2.1 and page 6, line 189 for section 2.2. We decided not remove sections 3.1 page 6 on line 221) because that section is the part of Design Thinking (DT) process, which is mandatory. This section talks about Empathize phase which is the process to found what problem is exist. On this work, We found problems experienced by Indonesia Regional Drinking Water Company (PDAM). Later, in the next DT phase called “Define” (section 3.2 at page 8 line 256), We propose a solutions to solve the problem found in the previous phase and the system explained on our manuscript. Then, in Ideate phase named “Ideate” (section 3.3 page 8, and line 267), we gather potential solution to solve problem faced by PDAM. In the next phase called “Prototyping” (section 3.4, page 12, line 370), We develop and creating the prototype of the companion mobile app. The last phase “Test” (section 4. Discussion, page 15 line 443), we test the prototype to make sure all things is working correctly.
Comments 3: A 2% error could be huge depending on the application. What is the error of the current method providing? Is 2% a great improvement? What about the performance of other designs? How does the proposed design compare to them? Response 3: Thank you for pointing this out. We agree with this comment. Therefore, We have revised the manuscript with more detailed explanation about the error rate and can be found at page 17, line 510-514. We also revised the manuscript to include peformance with other design by comparing the average reading error that can be found at page 17, lines 520-522.
Comments 4: There are a lot of writing style and format issues. Examples include unexplained abbreviations, mistaken referencing to figures (Line 253 on Page 8), and improper use of figures (Figure 19 provides no information), etc. Response 4: Thank you for pointing this out. We agree with this comment. Therefore, We have revised the manuscript to fix writing styles, formatting issues and include explanations of some abbreviations. We also revised the manuscript to fix mistaken referencing to figures that can be found at page 9, line 291 and removed improper use of figures. In addition, we utilize Quillbot Premium to assist us in identifying any grammatical errors.
|
||
4. Response to Comments on the Quality of English Language |
||
Point 1: Please see the Comments and Suggestions above. |
||
Response 1: Thank You for pointing this. We agree with this comment and revise all writing and format issues. We also revised the manuscript to add abbreviation explanation. |
||
5. Additional clarifications |
||
We have addressed all feedback from Reviewer 2, , as detailed in the point-by-point comments and suggestion above. |
Response to Reviewer 3 Comments
|
||
1. Summary |
|
|
Thank you very much for taking the time to review this manuscript. Please find the detailed responses below and the corresponding revisions and corrections in the re-submitted files.
|
||
2. Questions for General Evaluation |
Reviewer’s Evaluation |
Response and Revisions |
Does the introduction provide sufficient background and include all relevant references? |
Must be improved |
We have addressed this feedback, and the revisions can be found on page 2, lines 63-64.. |
Is the research design appropriate? |
Must be improved |
We have addressed this feedback, and the revisions can be found on page 5, lines 174-178. |
Are the methods adequately described? |
Must be improved |
We have addressed this feedback, and the revisions can be found on page 6, line 204. |
Are the results clearly presented? |
Must be improved |
We have addressed this feedback, and the revisions can be found on page 17-21, section 4.4, 4.5, and 4.6. |
Are the conclusions supported by the results? |
Must be improved |
We have addressed this feedback, and the revisions can be found on page 21, line 673-674. |
3. Point-by-point response to Comments and Suggestions for Authors |
||
Comments 1: A novel prepaid water meter system is proposed to monitor water consumption accurately compared to conventional meters. The LC sensors, Bluetooth low energy connectivity, tokens-based top-up system and mobile application are comprised in the prepaid water meter. The results show that the LC sensors is with precise results in measuring water usage. The paper seems organized well. It’s better to describe the novelty of this study clearly. Some grammar mistakes and awkward sentences preventing a proper understanding of the content are found. The following problems should be addressed. |
||
Response 1: Thank you for pointing this out. We agree with this comment. Therefore, We have revised the manuscript to include further explanation about novelty of the paper that can be found on Page 5, lines 174-188.
|
||
4. Response to Comments on the Quality of English Language |
||
Point 1: Extensive editing of English language required. |
||
Response 1: Thank you for your feedback regarding the clarity of the English in our manuscript. We sincerely apologize for any confusion caused by the language in the original submission. To address this issue, we have thoroughly revised the manuscript to improve the clarity and readability of the text. We hope that these revisions have made the content more accessible and easier to understand. If there are specific sections that remain unclear, we would appreciate your guidance so we can make further improvements. |
||
5. Additional clarifications |
||
We have addressed all feedback from Reviewer 3 as detailed in the point-by-point comments and suggestion above. |

Reviewer 2 Report
Comments and Suggestions for Authors
The authors of the manuscript set themselves the goal of developing a mobile application connected to a server to use tokenization for paying water bills in a prepaid manner. The developed solution aims to minimize social problems and reduce the costs of billing reading.
The authors of the manuscript used LC sensors for automatic reading of water meters and communication between the mobile application and the PWM device that controls the valve and reads the water meter readings.
The main part of the work consists in presenting the method of analyzing scientific articles containing keywords selected by the authors. To the reviewer's knowledge, preapid water meter systems come in many configurations. The use of LC sensors is also not innovative. Remote reading systems for water meters from leading manufacturers are based on LC sensors and magnetic induction (ITRON, Madallena, APATOR) or the Hall effect (Diehl Metering).
The problem of costs and savings resulting from the use of PWM devices, raised by the authors, has not been properly documented and proven.
The manuscript requires completion. After completing and responding to detailed comments, the manuscript will probably be able to be published.
Detailed comments:
1. In the Introduction part, you should present the water consumption in India, how much water costs, and what are the possibilities of purchasing water with an average income per person.
2. It should be explained why it is so important to develop a new prepaid device with mobile app?
3. What are the costs of processing payments after receiving the invoice (water)? Are remote reading systems used, e.g. radio, via GSM, LoRAWAN or other IoT?
4. It should be described how the process of purchasing water takes place (who and how reads the water meter readings) or paying the water fee (how does the recipient pay the fees?
5. In 3. Results, the authors wrote that the survey results indicate that the biggest problem is "PDAM Payment". How do customers pay for water? Will mobile payments in the application be more convenient, e.g. for older people? Does everyone have access to devices using the mobile application?
The description (1-5) should be included in the Induction part to outline the context of the authors' research and demonstrate the necessity of conducting it and developing PWM.
6. Are water meters in Indonesia only made by one manufacturer and one model? Does every water meter have a metal dial and is it located in the same place?
7. Is the solenoid valve powered by PWM or does it have a separate source? If it is powered by electricity from the mains, wouldn't it be better for the PWM to also be powered from the mains?
The description (6-7) should be included in the 3.3. Ideate
8. What dictates the change in battery life from 5 to 6 years? If it is for economic reasons, would it be justified to use 2 lithium batteries in the housing? Why was only one battery designed? Is it possible to increase the battery capacity? Many solutions for remote reading of water meters use lithium batteries with a capacity of 14-17 Ah.
9. Did the authors take into account the number of water token purchases? The article states that the device can perform 95 transactions within 5-6 years. What is the water purchasing trend in India? Do water consumers currently pay for water once a month, once a year, etc.?
10. There is no analysis of the device's lifespan in the case of purchasing individual tokens, e.g. purchasing a token every 1 m3. The device will then allow the recipient to purchase 95 m3 of water. How long will it take for an average consumer (family) to use this volume of water?
The description (8-10) should be included in the 4.4 Battery Usage Analysis
11. Part 4.5 of Cost Volume Profit Analysis should present a comparison of the current costs of measuring water consumption by currently connected water meters, the costs of the water meter reading process and the invoicing process in relation to the costs of purchasing the PWM solution and the costs of accounting services and mobile/bank payment fees, as well as the costs of system changes. IT of water and sewage companies necessary to implement such payments.
12. Part 4.3 LC Sensors Accuracy Testing (441-449). The measurement error concerns the error of the water measuring device (water meter). In the chapter, from what I understood, the authors noted a 2% error between the indication in the application/PWM and the indication from the water meter. These are additional apparent losses and a new problem in the water balance of the water utility. Please explain.
13. Please explain. Table 2. SLR publications evaluation result, does the communication method concern communication between the device controlling - the monitoring/payment system? The proposed PWM-mobile app - server system has two communication channels, which is worth explaining and writing down in a table. BLE is only between the application and PWM.
14. Part 5. Conclusions . What are the limitations in using the solution? Add information to manuscript.
Author Response
For research article
Response to Reviewer 2 Comments
|
||
1. Summary |
|
|
Thank you very much for taking the time to review this manuscript. Please find the detailed responses below and the corresponding revisions and corrections in the re-submitted files.
|
||
2. Questions for General Evaluation |
Reviewer’s Evaluation |
Response and Revisions |
Does the introduction provide sufficient background and include all relevant references? |
Must be improved |
We have addressed this feedback, and the revisions can be found on point-by-point response. |
Is the research design appropriate? |
Can be improved |
We have addressed this feedback, and the revisions can be found on point-by-point response. |
Are the methods adequately described? |
Can be improved |
We have addressed this feedback, and the revisions can be found on point-by-point response. |
Are the results clearly presented? |
Can be improved |
We have addressed this feedback, and the revisions can be found on point-by-point response. |
Are the conclusions supported by the results? |
Yes |
The Reviewer 1 has confirmed that the conclusions supported by the results. |
3. Point-by-point response to Comments and Suggestions for Authors |
||
Comments 1: In the Introduction part, you should present the water consumption in India, how much water costs, and what are the possibilities of purchasing water with an average income per person. |
||
Response 1: Thank you for pointing this out. We agree with your comment and have revised the manuscript to clarify the details related to water consumption in Indonesia (not India), including water costs and the average income of water customers to pay for it. These revisions can be found on page 2, lines 53-61. |
||
Comments 2: It should be explained why it is so important to develop a new prepaid device with mobile app? |
||
Response 2: Thank you for pointing this out. We agree with this comment and have revised the manuscript to emphasize this. The development of prepaid water meter with mobile app is a solution to address issues experienced by PDAM and their customers. This explanation can be found on page 2, lines 45 –52. |
||
Comments 3: What are the costs of processing payments after receiving the invoice (water)? Are remote reading systems used, e.g. radio, via GSM, LoRAWAN or other IoT? |
||
Response 3: Thank you for pointing this out. We agree with this comment and have revised the manuscript to explain more about this. This change can be found on page 2, lines 69-73. |
||
Comments 4: It should be described how the process of purchasing water takes place (who and how reads the water meter readings) or paying the water fee (how does the recipient pay the fees? |
||
Response 4: Thank you for pointing this out. We agree with your comment and have revised the manuscript to provide a more detailed explanation. These changes can be found on page 2, line 89, to page 3, line 106, and in Figure 1. |
||
Comments 5: In 3. Results, the authors wrote that the survey results indicate that the biggest problem is "PDAM Payment". How do customers pay for water? Will mobile payments in the application be more convenient, e.g. for older people? Does everyone have access to devices using the mobile application? |
||
Response 5: Thank you for pointing this out. We agree with your comment and have revised the manuscript to provide a more detailed explanation, including a comparison between the current payment system and the proposed payment system, which can be found on page 2, line 89, to page 3, line 106, and in Figure 1. Additionally, we have included an explanation regarding customer issues with 'PDAM Payment' on page 7, lines 247-248, to explain the problems experienced by PDAM customers. Furthermore, mobile payments have become commonplace in Indonesia with the rise of e-commerce, making mobile devices widely accessible across various segments of society. However, some water customers may still lack access to mobile devices. To accommodate this, a Point of Sales (POS) system was also developed for the PWM, but it is not discussed in this research as it falls outside the scope. Therefore, payment using mobile methods will be more convenient. We also conducted UI and UX testing of the mobile application to assess its suitability for PDAM customers, which can be found on page 15, line 463, to page 16, line 473, in section 4.1, 'Front-End Testing'. |
||
Comments 6: Are water meters in Indonesia only made by one manufacturer and one model? Does every water meter have a metal dial and is it located in the same place? |
||
Response 6: We agree with your comment and have revised the manuscript to provide a more detailed explanation. Water meters in Indonesia are produced by various manufacturers, all adhering to a single standardization known as Standar Nasional Indonesia (SNI), which dictates how water meters are made, including regulations for metal dials. This revision can be found on page 8, lines 268-274 and page 9, lines 281-283. |
||
Comments 7: Is the solenoid valve powered by PWM or does it have a separate source? If it is powered by electricity from the mains, wouldn't it be better for the PWM to also be powered from the mains? |
||
Response 7: We agree with your comment and have revised the manuscript to provide a more detailed explanation. The solenoid valve is powered by a lithium batteries for ease of implementation in the field and avoid intervention from outside parties. This revision can be found on page 9, lines 298-302. |
||
Comments 8: What dictates the change in battery life from 5 to 6 years? If it is for economic reasons, would it be justified to use 2 lithium batteries in the housing? Why was only one battery designed? Is it possible to increase the battery capacity? Many solutions for remote reading of water meters use lithium batteries with a capacity of 14-17 Ah. |
||
Response 8: Thank you for pointing this out. We agree with your comment and have revised the manuscript to provide a more detailed explanation. Due to the regulation that requires water meter retesting every 5 years, the PWM in this study was tested with an additional threshold for 6 years of use. We used only one battery a 3.6V 8500mAh because we want to design a small and compact PWM. Despite this, it remains functional for a period of 5 years. This changes can be found on page 18, lines 539-546. |
||
Comments 9: Did the authors take into account the number of water token purchases? The article states that the device can perform 95 transactions within 5-6 years. What is the water purchasing trend in India? Do water consumers currently pay for water once a month, once a year, etc.? |
||
Response 9: Thank you for pointing this out. We agree with your comment and have revised the manuscript to provide a more detailed explanation. The number 95 does not refer to transactions over 5-6 years; rather, it represents the number of times the PWM enters a deep sleep state, which contributes to the battery's 5-year lifespan. We extended this period to 6 years to include a one-year tolerance, as explained on page 18, lines 541-542. Currently, Indonesia operates on a postpaid system, where customers pay their water bills once a month at the end of their usage period. These clarifications can be found on page 19, lines 575-583, which detail the significance of the number 95, and on page 20, lines 597-606, which outline the water billing system in Indonesia and how customers pay their water bills. |
||
Comments 10: There is no analysis of the device's lifespan in the case of purchasing individual tokens, e.g. purchasing a token every 1 m3. The device will then allow the recipient to purchase 95 m3 of water. How long will it take for an average consumer (family) to use this volume of water? |
||
Response 10: Thank you for highlighting this point. We agree with your comment and have revised the manuscript to provide a more detailed explanation regarding the device lifespan based on the water consumption in liters per person per day (lpd) in Indonesia. These changes can be found on page 19, line 597, to page 20, line 606. |
||
Comments 11 Part 4.5 of Cost Volume Profit Analysis should present a comparison of the current costs of measuring water consumption by currently connected water meters, the costs of the water meter reading process and the invoicing process in relation to the costs of purchasing the PWM solution and the costs of accounting services and mobile/bank payment fees, as well as the costs of system changes. IT of water and sewage companies necessary to implement such payments. |
||
Response 11: Thank you for highlighting this point. We agree with your comment and have revised the manuscript to include a cost comparison between the current postpaid system and the proposed prepaid system for PDAM. These changes can be found on page 20, line 630, to page 21, line 634. |
||
Comments 12 Part 4.3 LC Sensors Accuracy Testing (441-449). The measurement error concerns the error of the water measuring device (water meter). In the chapter, from what I understood, the authors noted a 2% error between the indication in the application/PWM and the indication from the water meter. These are additional apparent losses and a new problem in the water balance of the water utility. Please explain. |
||
Response 12: Thank you for highlighting this point. We agree with your comment and have revised the manuscript to provide a more detailed explanation regarding error reading. These changes can be found on page 17, lines 512-514. |
||
Comments 13 Please explain. Table 2. SLR publications evaluation result, does the communication method concern communication between the device controlling - the monitoring/payment system? The proposed PWM-mobile app - server system has two communication channels, which is worth explaining and writing down in a table. BLE is only between the application and PWM. |
||
Response 13: Thank you for highlighting this point. We agree with your comment and have revised the manuscript to provide a more detailed explanation of the dual communication method used in this study. These changes can be found on page 5, lines 179-188. |
||
Comments 14 Part 5. Conclusions . What are the limitations in using the solution? Add information to manuscript. |
||
Response 14: Thank you for highlighting this point. We agree with your comment and have revised the manuscript to explain more about limitations of PWM in this work. These changes can be found on page 22, lines 693-698. |
||
4. Response to Comments on the Quality of English Language |
||
Point 1: - |
||
Response 1: Reviewer 2 mentioned that they are not qualified to assess the quality of English in this paper. To ensure the manuscript meets the necessary language standards, we have thoroughly reviewed and revised the text for clarity and correctness. |
||
5. Additional clarifications |
||
We have addressed all feedback from Reviewer 2, as detailed in the point-by-point comments and suggestion above. |
